# Nature-Based Solutions for Restoring an Agricultural Area Contaminated by an Oil Spill

**DOI:** 10.3390/plants11172250

**Published:** 2022-08-30

**Authors:** Elisabetta Franchi, Anna Cardaci, Ilaria Pietrini, Danilo Fusini, Alessandro Conte, Alessandra De Folly D’Auris, Martina Grifoni, Francesca Pedron, Meri Barbafieri, Gianniantonio Petruzzelli, Marco Vocciante

**Affiliations:** 1Eni S.p.A, Research & Development, Environmental & Biological Laboratories, Via Maritano 26, 20097 S. Donato Milanese, Italy; 2Institute of Research on Terrestrial Ecosystem, National Council of Research, Via Moruzzi 1, 56124 Pisa, Italy; 3Department of Chemistry and Industrial Chemistry, Università Degli Studi di Genova, Via Dodecaneso 31, 16146 Genova, Italy

**Keywords:** hydrocarbon biodegradation, phytoremediation, plant growth promoting bacteria, ecosystem restoration, next-generation sequencing, farming area

## Abstract

A feasibility study is presented for a bioremediation intervention to restore agricultural activity in a field hit by a diesel oil spill from an oil pipeline. The analysis of the real contaminated soil was conducted following two approaches. The first concerned the assessment of the biodegradative capacity of the indigenous microbial community through laboratory-scale experimentation with different treatments (natural attenuation, landfarming, landfarming + bioaugmentation). The second consisted of testing the effectiveness of phytoremediation with three plant species: *Zea mays* (corn), *Lupinus albus* (lupine) and *Medicago sativa* (alfalfa). With the first approach, after 180 days, the different treatments led to biodegradation percentages between 83 and 96% for linear hydrocarbons and between 76 and 83% for branched ones. In case of contamination by petroleum products, the main action of plants is to favor the degradation of hydrocarbons in the soil by stimulating microbial activity thanks to root exudates. The results obtained in this experiment confirm that the presence of plants favors a decrease in the hydrocarbon content, resulting in an improved degradation of up to 18% compared with non-vegetated soils. The addition of plant growth-promoting bacteria (PGPB) isolated from the contaminated soil also promoted the growth of the tested plants. In particular, an increase in biomass of over 50% was found for lupine. Finally, the metagenomic analysis of the contaminated soil allowed for evaluating the evolution of the composition of the microbial communities during the experimentation, with a focus on hydrocarbon- oxidizing bacteria.

## 1. Introduction

Oil spill refers to any accidental or intentional release of liquid hydrocarbons into the environment. The growing attention of society and governments to environmental issues has helped to reduce the number of large oil spills over the years [1]. However, even minor spills (e.g., connected to failures of pipelines) can give rise to severe accident scenarios and environmental degradation linked to the displacement of contaminants in the soil [2,3], with transboundary effects in case of watercourses or international seas, as recently demonstrated by [4].

Numerous approaches have been developed to address oil spills, ranging from chemical to biological, physical, and thermal methods [5]. Among the physical interventions, sorbents in particular are of interest as they can be used as (passive) containment or for the (active) removal of contaminants and can be particularly effective in recovering traces of oil from both land and water [6].

Indeed, adsorption is a simple and effective method for removing a wide variety of contaminants from aqueous solutions, both organic [7,8,9] and inorganic [10,11]. The further possibility of exploiting low-cost adsorbent materials from waste [12] represents a dual benefit for the environment, in line with the European dictates of the circular economy and the “near-zero discharge” of hazardous waste (which tends to virtually eliminate waste and possible consequent contamination through recycling and complete recovery of resources; see, for example [13,14]). 

These aspects have recently attracted considerable attention to the use of adsorbent materials for the management of oil spills threatening water bodies, also due to their potential capacity to recover oil and the lack of secondary pollution (if removed after use) [15]. In the case of terrestrial oil spills, the use of adsorbents is less frequent, but other solutions are possible [16,17]. However, an often-overlooked aspect is that remediation activities also have an environmental impact as they possibly exploit chemical products or processes, with consequent raw material and energy consumption. This approach can compromise the sustainability of the intervention itself [18], for example by generating harmful side effects such as the production of unwanted toxic residues [19], which can cause unexpected secondary contamination.

Remediation technologies based on nature-based solutions (NBS) represent an ecological and sustainable alternative that allows not only eliminating or reducing contamination but also minimizing the environmental impacts. Among the NBS remediation measures, a growing focus is on phytoremediation [20,21], which is the set of remediation technologies that see plants as the main actors in cleaning up organic and inorganic contaminants in soil and other environmental matrices (sediments, water). The interest in these phytotechnologies has grown over time thanks to their low cost, simplicity of operation and ecological benefits [22,23], and possible uses in combination with other solutions have also been proposed for further increasing overall sustainability [24,25]. In this regard, the use of green roofs [26] can improve the energy and environmental performance of urban environments [27,28] by combining, in urban areas, constructed wetlands techniques with other solutions to mitigate the progressive intensification of climate change. Different phytoremediation methodologies can be suitably exploited according to the type of contamination to be treated. For instance, in the case of contamination by potentially toxic metals, phytoextraction is considered a non-invasive approach to facing the problem in an ecological and economical way, removing the metals by adsorption from the roots and accumulation by translocation in the different tissues of the plant [29]. 

However, plants can absorb only those species present in soluble form in the soil solution [30,31]. Strategies are constantly being explored to assist this process. For instance, chelating/mobilizing agents can promote the release of metal ions from the soil particles to the soil solution [32] increasing their bioavailability for absorption by plants, thus offering the possibility of using fast-growing and highly tolerant species instead of hyperaccumulating ones. [33,34]. Another possible strategy is the use of plant growth-promoting bacteria (PGPB), such as those that populate the rhizosphere (PGPR) [35] that support the plant in its biological activities, e.g., by modulating the production and level of phytohormones and by facilitating the bioavailability of soil nutrients [36]. In some cases, they can also promote the mobility and bioavailability of metals in the soil, increasing their absorption by plants [37,38,39].

In the case of oil spills, when petroleum hydrocarbons enter an uncontaminated environment, the change in that environment is almost immediate due to dispersion processes of the polluting organic molecules in the soil matrix, partial evaporation and absorption. After that, chemical modifications such as oxidation take over, and finally, the processes of catabolism take place [40]. The biodegradation process is mainly mediated by the autochthonous microbial community of the soil (natural attenuation). These microorganisms undergo sudden and significant qualitative and quantitative changes with the selection and enrichment of the potentially most valuable species for that specific contamination [41].

The addition of electron donors or acceptors and nutrients could significantly accelerate the biodegradation process. A metagenomic analysis [42] of the indigenous microbial community in the contaminated matrix is essential for stimulating the potentially more helpful microorganisms effectively. Understanding the interaction of anthropogenic contaminants with the soil and microbial communities is crucial in designing bioremediation interventions. The various hydrocarbon-oxidizing microorganisms often possess only some of the catabolic enzymes necessary to degrade the hydrocarbon molecules completely. Synergy with other microorganisms (consortia) is crucial, as only these symbiotic relationships allow them to fully exploit their effective potential [43,44]. 

In this context, the use of plants plays an essential role as it affects the biodegradation of petroleum hydrocarbons providing an optimum environment for the proliferation of the hydrocarbon-oxidizing microorganisms. This activity often leads to a more significant reduction of hydrocarbons in vegetated soils than in non-vegetated ones since rhizodegradation is the primary mechanism for the disappearance of petroleum hydrocarbons [36,45]. According to soil characteristics, contamination can also be reduced by the uptake of some plant species, where contaminants can be further degraded to harmless substances or immobilized through root uptake and accumulation [46]. 

In this study, a laboratory experiment was performed with the soil of an agricultural area where a break-in caused a substantial spill of diesel oil from an oil pipeline adjacent to the field. Two distinct approaches were adopted. The first involved the biodegradative capacity of the indigenous microbial community through laboratory-scale experimentation with different treatments (natural attenuation, landfarming, landfarming + bioaugmentation). The second consisted of evaluating the effectiveness of phytoremediation by testing the activity of three plant species: *Zea mays*, *Lupinus albus* and *Medicago sativa*, also supported with the inoculum of PGPB. This investigation was intended to verify the applicability of bioremediation interventions in the area to allow the complete restoration of the functionality of the impacted ecosystem.

## 2. Materials and Methods

### 2.1. Site Description and Sampling

The site under investigation is an agricultural area of about 10,000 m^2^ in northern Italy. A large quantity of diesel fuel was poured out due to a fuel burglary at an oil pipeline next to the field. For the activity described in this work, nine superficial and deep soil samples were taken in correspondence with previous survey points (Figure 1); for each point, about 25 kg of soil was taken for the laboratory tests and about 1 kg for chemical analyses. The soil was homogenized and sieved at 2 cm directly in the field. 

### 2.2. Soil Characterization

Before setting up the experiment, the soil samples were subjected to chemical analysis to assess the actual contamination. Table 1 lists the samples and the relative concentrations of the hydrocarbon fractions C ≤ 12 (EPA 5021A 2014 + EPA 8015D 2003) and C > 12 (EPA 3550C 2007 + EPA 8015D 2003), determined by GC-MS analysis. Table 1 also shows the values of volatile hydrocarbons (VOCs) recorded directly in the field during sampling.

The concentrations of detected hydrocarbons, higher for S3 (the one adjacent to the barrier) than for the other samples, guided the choice of samples for subsequent experiments. Sample S3 was chosen for the bioremediation approach, while all the other samples were used to explore the second approach (phytoremediation). In the latter case, a physical characterization was also conducted, determining the soil texture according to [47]. 

Table 2 shows the soil texture of the samples used for the phytoremediation test (all taken from the field except for S3).

### 2.3. Selection of the Hydrocarbon-Oxidizing Bacteria

First, a total microbial count was performed to verify the effective biodegradation potential of the soil. The method used to evaluate this parameter is that of serial dilutions: 5 grams of each soil sample were added to 45 mL of 0.1% (*w*/*v*) solution of sodium pyrophosphate. After homogenization for 30 min, this solution was decimally diluted (10^−1^ to 10^−7^), and aliquots of the resulting solutions were plated on an LB agar medium. After incubation at 30 °C for 5 days, the colony forming units (CFU) per gram were calculated.

The eight samples (BH3, C05, PZ7, C08, C41, PZ9, C13, PZ8) at the two depths and sample S3 taken from the barrier (the point adjacent to the break-in point) at the three depths were mixed. The composite sample thus obtained was used to select the hydrocarbon-oxidizing bacteria. Specifically, to one gram of this soil sample was added a solution of trace elements (MnCl_2_ ∙ 2 H_2_O 5 g L^−1^, ZnCl_2_ 3 g L^−1^, CuCl_2_ ∙ 2 H_2_O 0.9 g L^−1^, CoCl_2_ ∙ 6 H_2_O 1 g L^−1^, Na_2_MoO_4_ ∙ 2 H_2_O 1 g L^−1^, NiCl_2_ ∙ 6 H_2_O 0.3 g L^−1^, H_3_BO_3_ 3 g L^−1^, Na_2_O_3_Se ∙ 5 H_2_O 0.2 g L^−1^, pH 1), MEM Vitamin Solution (Merck^®^) and 5% of diesel oil as the only source of carbon, for a total volume of 200 mL. This suspension was divided into 4 Erlenmeyer flasks of 250 mL (with 50 mL each), which were then incubated at 30 °C with shaking. The suspension showed microbial growth after three days of incubation, as indicated by colonized diesel droplets. Cultures were then diluted at 10% and reincubated for the other 3 days, repeating this step 2 more times. The bacterial culture from the final enrichment step was then washed to eliminate the added diesel and used for the bioaugmentation phase of the first approach. About 10^11^ CFU of this bacterial culture was added to the trays labelled landfarming + bioaugmentation. 

### 2.4. Characterization of the Selected Hydrocarbon-Oxidizing Bacteria 

The enrichment of the hydrocarbon-oxidizing bacteria was then used to isolate and characterize the microorganisms. Serial dilutions (10^5^–10^7^) of the suspension were prepared in sterile water and then plated on LB and R2A agar and incubated at 30 °C for 5 days. Several colonies were visualized but with a few different phenotypes, consistent with a recent contamination. After a few passages of plate streaking, 20 pure colonies were selected. Genomic DNA was extracted with a Maxwell 16 system (Promega^®^), and the 16S rRNA fragments were amplified via polymerase chain reactions (PCR) [48]. 

A BigDye^®^ Terminator v3.1 Cycle Sequencing Kit (Applied Biosystems Inc. 850 Lincoln Centre Drive Foster City, CA 94404 USA) and the same primers used in the previous PCR were used to sequence the PCR-amplified 16S rRNA with an automated DNA sequencer (SEQ Studio Genetic analyzer).

A SeqMan application from Lasergene v. 16.0.0 (DNASTAR, Inc.3801 Regent St.Madison, WI 53705 USA)was employed to edit and assemble the nucleotide sequences obtained. Then, a homology comparison was performed (basic local alignment search tool, BLAST, analysis), exploiting the information available at the National Center for Biotechnology Information server (NCBI, www.ncbi.nlm.nih.gov/blast/Blast.cgi, accessed on 18 February 2022) [49].

### 2.5. Setting Up the Biodegradation Experimentation

This experiment, conducted exclusively on the S3 sample, was performed to evaluate on a laboratory scale the possibility of carrying out a bioremediation intervention by exploiting the biodegradation capacity of the indigenous microbial flora. Sample S3 from the barrier was dried under a hood for 4 days and then sieved up to 2 mm. Then, 6 polypropylene trays (3 treatments in duplicate) of 30 × 20 × 13 cm (l × w × h), each containing 1 kg of soil, were prepared. The three treatments [50] were as follows:NA (natural attenuation): weekly moistening.L (landfarming): weekly moistening + mixing.LB (landfarming + bioaugmentation): weekly moistening + mixing + microbial inoculation at the initial time.

In all cases, tap water was regularly added to maintain 10–20% moisture. As specified in Section 2.3, for the bioaugmentation phase, the enrichment suspension obtained after washing the soil to eliminate the added diesel was used as the inoculum (that is, before the isolation and characterization of microorganisms). At times 30′, 60′, 90′, 120′ and 180′, composite samples were taken from each tray and subjected to chemical analysis to monitor biodegradation.

### 2.6. GC-MS Analysis of Samples

Ten grams of each sample dispersed in quartz with pre-bed in diatomaceous earth were extracted using an automatic Buchi Speed Extractor E-916 extractor. The solid matrix extraction procedure was specifically developed and optimized to respond to the particular characteristics of the matrix. Hexane-dichloromethane 9:1 *v*/*v* was used as extracting solvent. The extraction was carried out at a temperature of 50 °C and a pressure of 50 bar in a 20 mL steel cell for a hold time of 5 min, followed by washing with solvent for 3 min and washing with nitrogen for the same time interval, all repeated for 10 extraction cycles. The extracts were subsequently concentrated to 10 mL via a heated nitrogen flow concentrator and bath and subsequently concentrated and diluted.

The analysis of extracts was performed using an Agilent 7820A GC connected to an MSD5977E mass selection detector with a split/splitless injector (operating in split mode) was used, equipped with an Agilent HP5 MS 30 m, 0.25 mm, 0.25 mm column, and a thermal gradient of 40 °C (isothermal for 2 min); ramp of 7 °C/min up to 270 °C; ramp of 15 °C/min up to 320 °C; isotherm of 10 min. The acquisition was carried out in “SCAN” mode, with the full scan of the MS spectrum from mass 50 to 600. For quantifying the identified compounds, a calibration curve with 5 points with a concentration range of 5–100 ppm was used, with a control point (outside the curve and not used for analyses) of linearity at 200 ppm. 

### 2.7. In Vitro Assessment of PGP Properties and Inoculum Preparation

The two strains at biohazard level 1 (CG13, CG16) isolated from the soil were subjected to a series of in vitro assays to evaluate their potential to promote plant growth. In detail, indole acetic acid (IAA) production was estimated according to the protocol of Shahab et al. [51]. The ability of the isolates to solubilize inorganic phosphate was determined by growth in NBRIP (National Botanical Research Institute’s phosphate growth medium): 10 g L^−1^ glucose, 5 g L^−1^ Ca_3_ (PO_4_), 2.5 g L^−1^ MgCl_2_ 6H_2_O, 0.25 g L^−1^ MgSO_4_ 7H_2_O, 0.2 g L^−1^ KCl and 0.1 g L^−1^ (NH_4_)_2_SO_4_, supplemented with 0.025 g L-1 bromophenol blue (BPB) according to the method developed by Nautiyal [52]. On the other hand, the production of siderophores was evaluated according to the method described by Milagres et al. [53]. The production of exopolysaccharides (EPS) was estimated as described by Santaella et al. [54] using the following medium: 0.1 g L^−1^ MgSO_4_ 7H_2_O, 0.1 g L^−1^ CaCl_2_ 2H_2_O, 0.022 g L^−1^ FeSO_4_ 7H_2_O, 0.02 g L^−1^ EDTA, 0.43 mg L^−1^ ZnSO_4_, 1.30 mg L^−1^ MnSO_4_, 0.75 mg L^−1^ NaMoO_4_ 2H_2_O, 2.80 mg L^−1^ H_3_BO_3_, 26 μg L^−1^ CuSO_4_ 5H_2_O, 70 μg L^−1^ CoSO_4_ 7H_2_O, 7.9 g L^−1^ K_2_HPO_4_, 7.5 g L^−1^ KH_2_PO_4_, 0.1 g L^−1^ yeast extract and 20 g L^−1^ sucrose. Isolate strains were thereby tested for ammonia production by culturing them in peptone water, 5 g L^−1^ peptone and 5% NaCl, pH 7.2 for 72 h at 30°C. Nessler’s reagent (0.5 mL) was then added to each tube. The development of the yellow-brownish colour indicated NH_3_ production. The proteolytic activity (casein degradation) was determined as described by Nielsen and Sørensen [55]. The positive isolates showed a clearing zone in milk skim agar (7% skim milk, 5 g L^−1^ casein, 1 g L^−1^ glucose, 2.5 g L^−1^ yeast extract and 15 g L^−1^ agar) after four days of incubation at 30°C. The isolates were also tested for their ability to form biofilms in vitro (film) by inoculating them in glass tubes with 7 mL of LB (Luria Bertani) medium. The tubes were incubated at 30°C for seven days without shaking. A visible layer (film) formed at the interface between medium and air indicated a potential ability to produce biofilms. For the inoculum preparation, the two strains were grown in LB medium for 48 h. Then the cell pellets obtained by centrifugation (9000 rpm, 20’) were combined and resuspended with a solution composed of 1% sodium glutamate and 7% sucrose. The suspension was divided into small aliquots containing enough CFU to be used directly in the phytoremediation test pots, calculating about 108 CFU g^−1^ of soil. Aliquots were frozen for 16 h and then lyophilized for storage until use.

### 2.8. Phytoremediation Trials: Micro- and Mesocosm Tests

Considering the high biological variability, experimentation with plants requires several replicates. Thus, it was necessary to organize the investigation so that the tested samples were as informative as possible. The soils were then grouped into four samples considering the concentration of contaminants and the sampling depth, as shown in Table 3.

Experiments were conducted following a completely randomized design at a microcosm scale, using 500 g of each contaminated soil for each of the three species, sown with 0.8 g of *Medicago Sativa* (alfalfa) seeds, 5 seeds of *Zea Mays* (corn) and 6 seeds of *Lupinus Albus* (lupine). Under the same experimental conditions, two control microcosms were set up for each species using uncontaminated agricultural land (control sample). In addition to these tests, an equal number of microcosms was set up with the same soils and plant species but with the addition of the PGPB inoculum to the soils. Five replicates (number of separate plant specimens) were made for each treatment, and the growths were performed in a growth chamber (CCL300BH-AS S.p.A., Perugia, Italy) with the following conditions: photoperiod of 14 h light at 24 °C and 10 h dark at 19 °C, photosynthetic photon flux density of 130 µmol m^−2^ s^−1^. All the microcosms were irrigated daily with tap water only as the soils of the contaminated agricultural area contained nitrogen, phosphorus and potassium in sufficient quantities. It was not necessary to add fertilizers. After 30 days, the plants and the soil were collected by separating the roots and shoots, and fresh weight (FW) was determined as described in [49]. 

Once the microcosm growth tests were completed, a greenhouse experiment (mesocosm) was set up with the plant species *Z. mays* to evaluate the effect of plants on biodegradation. More soil and a longer duration of the plant’s vegetative cycle were used for this test. Each mesocosm was prepared with 5 kg of soil (contaminated or of control), and 14 seeds of *Z. mays* were sown in each pot. The experiment, conducted with five replicates, followed the same experimental scheme as the microcosm test. All the mesocosms were periodically irrigated according to the needs of the plants, and water was also added to the non-vegetated mesocosms, thus imitating the processes of natural attenuation. After about 60 days from sowing, the plants were harvested, and the vegetated and non-vegetated soils were analyzed by measuring the concentration of the hydrocarbon fraction C > 12, as previously described.

### 2.9. Next-Generation Sequencing (NGS) Analysis_Ion Torrent Sequencing

A sample of 3 ng of the genomic DNA, obtained by the extraction of 500 mg of soil samples through the Fast DNA^®^ Spin Kit for Soil (MP Biomedicals) and quantified with a Qubit^®^ 2.0 fluorometer (Invitrogen), were amplified using the 16S Metagenomics Kit (Life Technologies, 5781 Van Allen Way Carlsbad, CA, USA 92008). The amplification program was set up as follows: 95 °C for 10 min, followed by 25 cycles at 95 °C per 30 s, 58 °C for 30 s and 72 °C for 20 s, with a final hold time for 7 min at 72 °C and a cooling step at 4 °C. The subsequent purification of the amplicons, the preparation and the sequencing of the libraries followed the protocols for the Ion GeneStudio S5 Systems (i.e., Ion Chef^TM^ System and Ion GeneStudio S5 Sequencer) provided by the manufacturer. The run is based on the workflow Metagenomics 16S w1.1 handling the Database Curated microSEQ^®^16 S and the reference Library 2013.1. The primers detected both ends to obtain 250 bp sequences. Alignment in Torrent Suite™ (v.5.16, Life Technologies Corporation | 200 Oyster Point Blvd | South San Francisco, CA 94080 | USA) is performed using the Torrent Mapping Alignment Program (TMAP). The sequences that occurred only once in the entire dataset were removed, and the representative sequences were defined with a 97% similarity cut-off. After classifying the operational taxonomic unit (OTU) representative sequences, the output was elaborated to obtain a relative abundance (%) of each OTU in the total amounts of the entire sample.

### 2.10. Statistical Analysis

All statistical analyses were performed using STATISTICA (v.6.0 by StatSoft Inc. 2300 E 14th St Tulsa OK Oklahoma United States 74104).

## 3. Results and Discussion

### 3.1. Assessment of the Soil Biodegradation Potential

Table 4 shows the microbial counts of all the soil samples. The evaluation of the cultivable species revealed that the CFUs were sufficient to indicate a probable ongoing biodegradation activity. The isolation and characterization of the indigenous hydrocarbon-oxidizing bacteria were then performed.

The subsequent selection of the hydrocarbon-oxidizing microorganisms through selective growth in a minimum culture medium with the addition of 5% diesel oil allowed for the isolation of 20 bacterial strains. Figure 2 lists the 20 isolates with their biohazard classifications.

### 3.2. Evaluation of the Biodegradation Effectiveness

The experiment in the laboratory showed that in just 60 days, the percentage of biodegradation was higher than 50% in all three treatments, namely NA = natural attenuation, L = landfarming and LB = landfarming + bioaugmentation (Figure 3).

At the end of the experiment, after 180 days, high percentages of reduction in the number of contaminants by biodegradation were detected: for the NA treatment, 86% for the C9–C18 fraction, 81% for the C19–C36 fraction, 93% for the linear hydrocarbon fraction and 76% for the branched ones; for the L treatment, 82% for C9–C18, 76% for C19–C36, and 83% for linear and branched. Finally, the LB treatment generally showed the best results with 91% biodegradation of C9–C18 hydrocarbons, 86% for C19–C36, 96% for linear and 82% for branched ones. Diverse percentages of degradation are detected concerning the different molecular groups analyzed. Although the numbers for each sample do not differ much, the more complex molecules are more recalcitrant to the enzymatic attack. Generally, their degradation occurs more slowly than with simpler molecules (linear and with fewer C atoms).

The results obtained assume the presence of an active hydrocarbon-oxidizing microbial community. The strains selected from the enrichment cultures with diesel oil as the only carbon source belong mainly to the genera *Bacillus* and *Burkholderia*. Even if the cultivable bacterial species represent a minimal percentage (<1%) of the microbial communities present in the soil, these genera are known producers of surfactant molecules. They are certainly a good indication of probable biodegradative activity.

Several examples support this hypothesis. For instance, two bacterial strains, *Bacillus cereus* T-04 and *Bacillus halotolerans* 1-1 [56], were isolated from soil contaminated with crude oil and proved to be highly efficient during a laboratory simulation. The inoculum of the two strains in contaminated soil samples allowed the biodegradation of the crude oil to reach, after 180 days of treatment, about 97.5%, compared with the untreated control samples, which stopped at 26.6%. 

Another study [57] analyzed the capacity of some strains, including *Burkholderia* sp., *Pseudomonas* sp. and *Cupriavidus* sp., to degrade binary mixtures of octane (as an aliphatic component) with benzene, toluene, ethylbenzene or xylene (BTEX, such as aromatic hydrocarbons). The *Burkholderia* strain degraded all BTEX compounds faster than octane. This result suggests that *Burkholderia* played a crucial role in the preferential degradation of aromatic hydrocarbons over aliphatic hydrocarbons. 

### 3.3. Metagenomic Analysis

Generally, the data show a hydrocarbon-degrading microbial community, mainly composed of the orders Acidomicrobiales [58,59], Actynomicetales [60], Bacillales [61,62], Pseudomonadales [63,64], Rhizobiales [65] Sphingobacteriales [58,59], Sphingomonadales [66,67] and Xanthomonadales [58,62] (Figure 4). 

Among these, the four orders highlighted in Figure 4 (i.e., Acidimicrobiales, Actinomycetales, Sphingobacteriales and Xhantomanadales) showed a significant shift during the different incubation times and conditions. In detail, after 90 days can be seen an increase in the order of Acidomicrobiales in all the samples (i.e., NA and treated), while the order of Sphingobacteriales is valuable only in the treated samples. This increase is mainly due to the Chitinophaga family that raises values of relative abundances equal to 17% and 18%, respectively, at T90 and T180 in the landfarming treatment (L). Similarly, 14% and 27% values are reached at T90 and T180, respectively, in the landfarming treatment with bioaugmentation (LB) (Figure 4).

The order Actynomicetales showed the highest increase in each condition after 90 and 180 days of incubation compared with the respective T30. This is mainly due to the increase in relative abundance values to around 11% of the Mycobacteriaceae family in all these samples. The samples belonging to the NA condition registered higher values (i.e., 28%) of the order Xanthomonadales at T30. Their relative abundance is mainly due to the family Xanthomonadaceae, which decreases during the incubation (i.e., 13% at T90 and 5% at T180), as in the other conditions studied. 

### 3.4. PGPB Inoculum Features

Only two of the twenty hydrocarbon-oxidizing strains isolated from contaminated soil were classified with biological hazard level 1 (although DSMZ has provisionally been ranked as level 2). These isolates were evaluated as potential PGPBs. Both strains (*Kocuria rhizophila* and *Bacillus wiedmannii*) were shown to possess three of the most interesting promoting properties: IAA, siderophores and ammonia production. *Bacillus weidmannii* was also positive for protease production and biofilm formation in vitro, and *Kocuria rhizophila* also showed the ability to solubilize inorganic phosphate (Figure 5). 

It was observed [68] that the application of *K. rhizophila* significantly influenced the growth promotion and metal uptake capacity of *Glycine max* L. in industrially contaminated soils. The use of *K. rhizophila* in association with citric acid led to an increase in plant biomass of approximately 38.73% compared with uninoculated plants. Additionally, a strain isolated from the *Zea mays* rhizosphere and identified as *Kocuria rhizophila* Y1 [69] tolerated up to 10% NaCl and showed two growth-promoting characteristics: phosphate solubilization and IAA production. The inoculation of *Z. mays* plants with this strain under salinity conditions significantly improved biomass production, photosynthetic capacity, antioxidant levels and chlorophyll accumulation compared with uninoculated plants. In a study performed by Saran et al. [70], six Pb- and Cd-tolerant PGP-tolerant bacterial strains were isolated and selected from the roots of the aromatic plant *Helianthus petiolaris*. Among these isolates, the strain *Bacillus wiedmanni* ST29 was inoculated in plants of *Helianthus annuus* and lowered the bioaccumulation of Cd by 40%.

### 3.5. Effect of PGPB on Biomass

One of the most evident effects of the action of PGPB is the improvement of the health of plants. Properties such as the increased availability of phosphorus, nitrogen and iron and the production of auxins such as IAA are beneficial actions allowing plants to better tolerate the abiotic stress caused by pollution. Among the positive effects, there is undoubtedly greater biomass: One of the essential parameters for the success of a phytotechnology intervention is greater biomass production.

The first growth tests in the microcosm in the four composite soils (B1, B2, B3 and B4) showed, especially in the aerial portion, lower percentages of fresh biomass compared with the relative growths in agricultural control soil (Table 5). There is growth of about 33% less for alfalfa, 38% for lupine and 40% for corn. On the other hand, Figure 6 shows the results obtained with the PGPB inoculum. The positive effect is significant, and there is an increase in biomass of up to 50%, especially in lupine (soil B4) and alfalfa (soils B1, B2). 

### 3.6. Effect of Plants on Biodegradation

The soil under study is in dynamic contamination conditions, which naturally reduces hydrocarbons poured into the field following the spill. This condition is undoubtedly very different from that found in a former industrial site where the presence of the same contaminants has reached equilibrium with the soil’s various biotic and abiotic components. In the latter case, the degradation processes have significantly slower kinetics. The fundamental action of the rhizosphere is expressed through the direct activity of the hydrocarbon-oxidizing bacteria by promoting the growth of plants in conditions of stress. Plant–microorganism interaction plays a crucial role in removing contaminants from the soil [71]. The main action that plants perform is to stimulate and promote the activity of hydrocarbon-oxidizing microorganisms, thanks to the release of radical exudates [45], although to a lesser extent, plants can also accomplish their action by absorbing contaminants and accumulating them at the root level [72]. Figure 7 shows the C > 12 hydrocarbon content variation in the mesocosm test performed with the species *Z. mays*.

These results indicate that the contribution of plants appears significant, highlighting an increase in degradation of 15–18% in vegetated soils compared with non-vegetated ones. Therefore, we can assume that the presence of plants always favors decreases in hydrocarbons in the soil. 

## 4. Conclusions

This feasibility study aimed at two main management objectives: removing the maximum amount of contaminants from the soils affected by the oil spill and recovering a correct agronomic practice to be carried out in absolute safety.

Regarding the phytoremediation experimentation, an expected decrease in biomass yield with the polluted soil was noted compared with the control soil. Still, the tested plants were able to grow satisfactorily in the soils under examination. At the end of the tests, a decrease in the concentration of hydrocarbons was observed, favored by the presence of plants. Indeed, the addition of the defined PGPB (plant growth promoting bacteria) microorganisms promoted growth on contaminated soils, and the production of fresh biomass was similar to that in the control medium. Therefore, organic contaminants can be concretely reduced by the joint action of plants and microorganisms using organic compounds as a primary carbon source. In particular, the rhizodegradation processes favor contaminants’ degradation with significant contamination reductions. The two approaches were applied separately in this first phase of the investigation. However, the final goal is to combine them since using integrated strategies (soil treatment to promote and accelerate biodegradation and the use of suitable plant species) can lead to the complete redevelopment of the area with an excellent probability of success. The positive response to the feasibility test allowed for the preparation of an operational remediation plan that provides a view to a complete restoration of the ecosystem services in the area. A series of technologies differentiated according to the level of contamination was planned. A field test is now underway on a total area of 600 m^2^: an enhanced bioremediation intervention is being applied (with oxygen and nutrient injections) on deep soils and a phytomediation intervention on superficial soils with *Zea mays*, *Sorghum vulgare*, *Ricinus communis* and *Helianthus annuus*. In the area most affected by the spill (the embankment), the provision of a reactive barrier with a product based on activated carbon and phyto-capping is foreseen.

## Figures and Tables

**Figure 1 plants-11-02250-f001:**
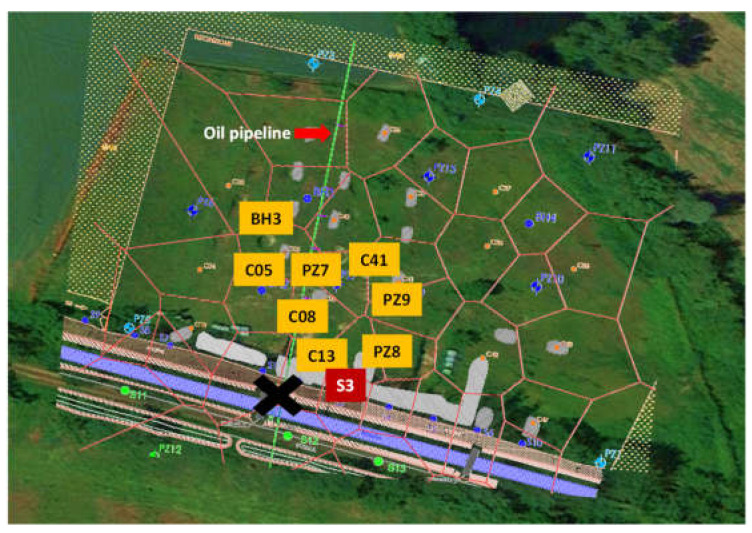
Map of the area under study with the indication of the sampling points, the trace of the oil pipeline and the break-in point (black cross).

**Figure 2 plants-11-02250-f002:**
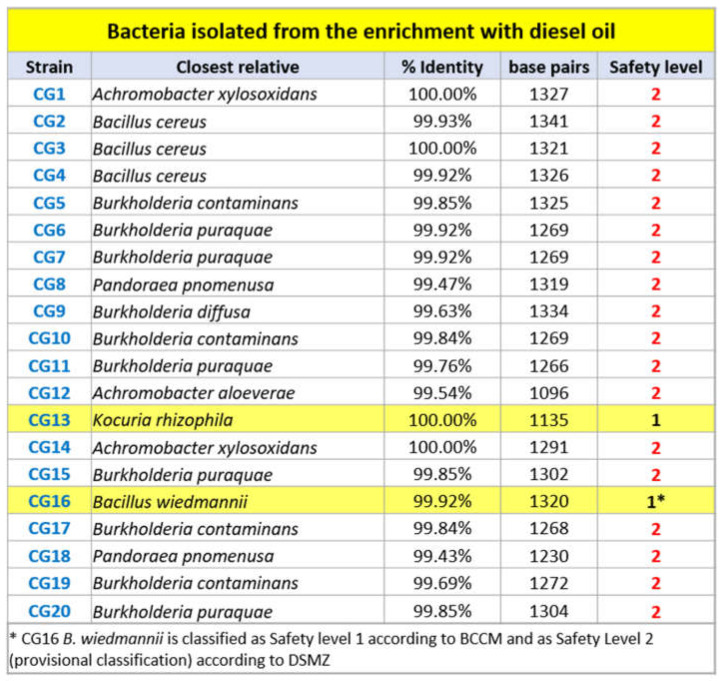
List of the 20 isolated strains. Biological safety levels (according to three different database: “Classification of Prokaryotes—Bacteria and Archaea—into Risk Groups”, TRBA 466; www.baua.de/abas, accessed on 20 February 2022; Leibniz Institute DSMZ, German Collection of Microorganisms and Cell Cultures GmbH https://www.dsmz.de/, accessed on 20 February 2022; BCCM: Belgian Coordinated Collections of Micro-organisms https://bccm.belspo.be/, accessed on 20 February 2022)) are also shown.

**Figure 3 plants-11-02250-f003:**
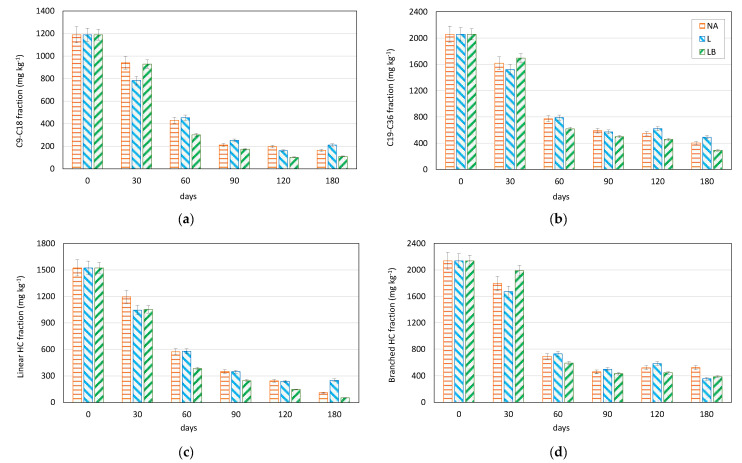
Evaluation of the residual content C9–C18 (**a**), C19–C36 (**b**), linear (**c**) and branched (**d**) fractions) after the different treatments (NA, L, LB) at the time of collection (30, 60, 90, 120, 180 days) in comparison to the initial content (t0). NA = natural attenuation, L = Landfarming, LB = landfarming + bioaugmentation.

**Figure 4 plants-11-02250-f004:**
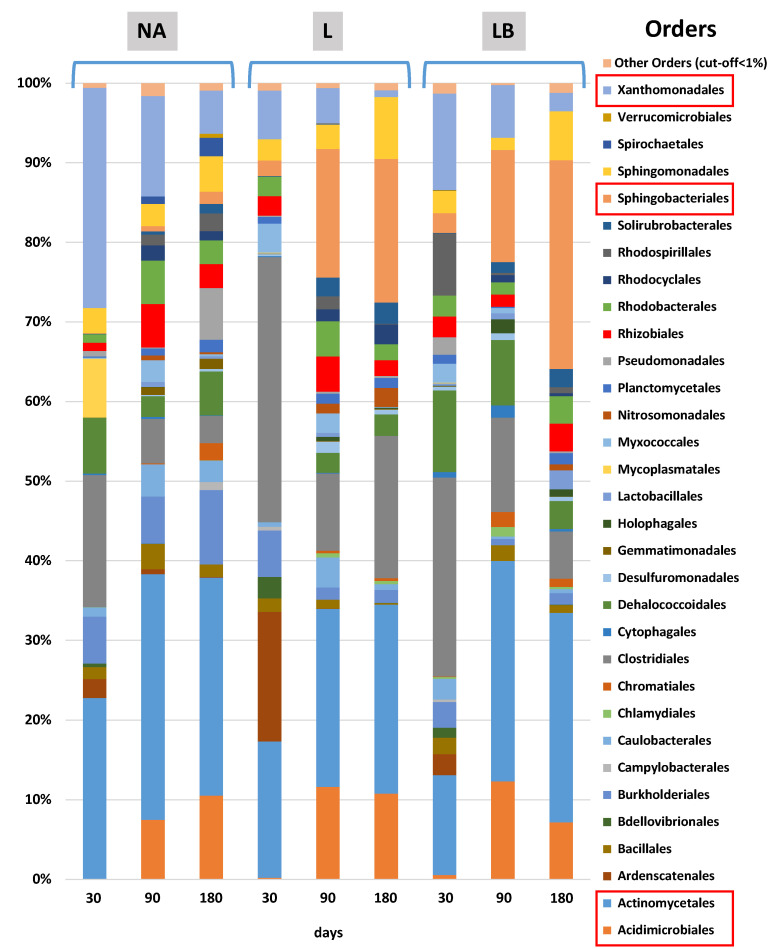
NGS Ion Torrent analysis of all the samples from the three treatments (NA, L, LB) at 30, 90 and 180 days. NA = natural attenuation, L = landfarming, LB = landfarming + bioaugmentation.

**Figure 5 plants-11-02250-f005:**
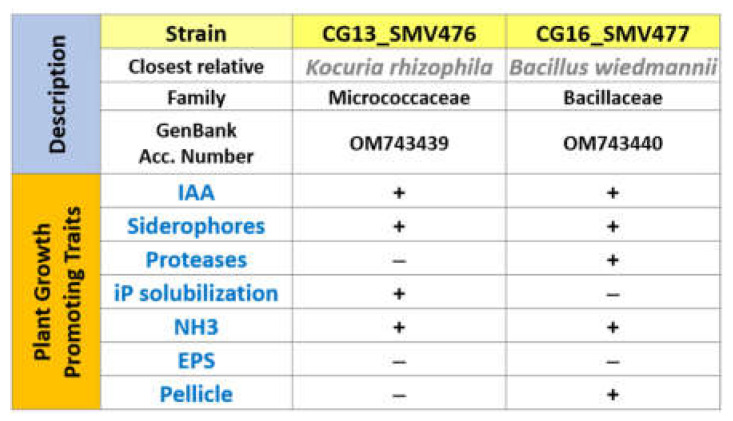
Description of the growth-promoting properties shown by the two strains CG13 and CG16 chosen as inoculum for the phytoremediation tests.

**Figure 6 plants-11-02250-f006:**
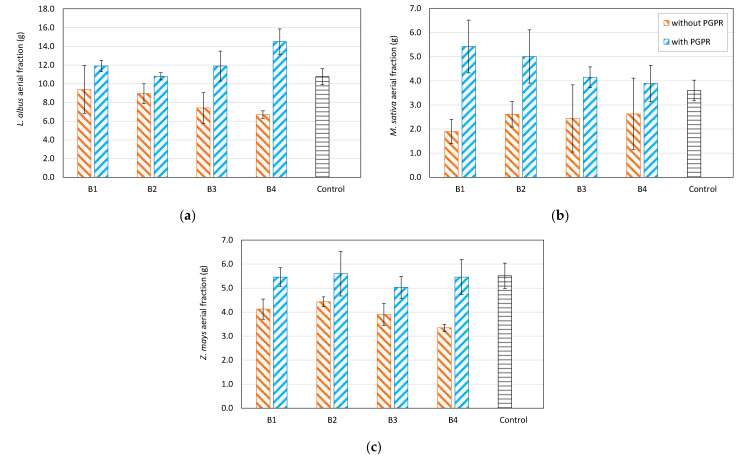
Comparative growth of the aerial part of Lupine (**a**), Alfalfa (**b**) and Corn (**c**) on the contaminated soils without (orange bars) and with (light blue bars) the inoculation of the two isolated and characterized PGPB strains with respect to control soil not contaminated (grey bar).

**Figure 7 plants-11-02250-f007:**
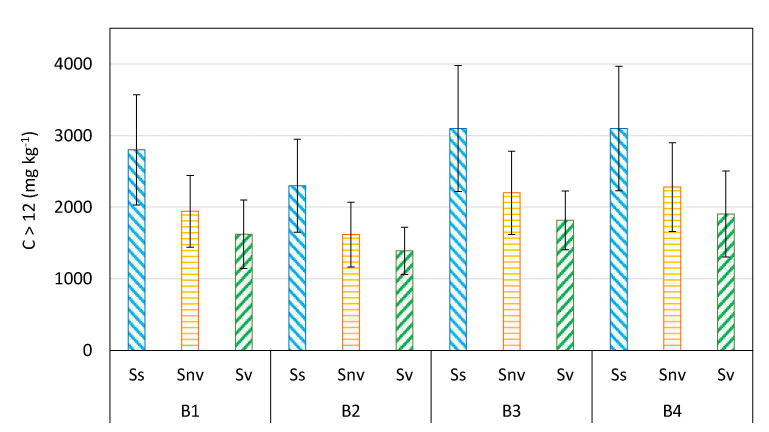
Concentration of the residual hydrocarbons in the four soils, B1, B2, B3 and B4, with the comparison between the starting soil (Ss, light blue bars), the non-vegetated soils (Snv, orange bars) and the vegetated ones (Sv, dark green bars) with *Z. mays*.

**Table 1 plants-11-02250-t001:** Hydrocarbon concentrations (volatile fraction, C ≤ 12 and C > 12) detected in the considered soil samples and depths at which the samples were taken.

Sample	Depth (m)	VOCs (mg kg^−1^)	C ≤ 12 (mg kg^−1^)	C > 12 (mg kg^−1^)
BH3	0–1	100	76	4500
	1–2	133	80	3800
C05	0–1	85	60	2200
	1–2	156	95	4900
PZ7	0–1	152	61	1900
	1–2	149	100	3600
C08	0–1	152	80	3900
	1–2	175	130	5200
C41	0–1	99	14	1200
	1–2	131	66	3100
PZ9	0–1	82	47	2300
	1–2	140	120	5100
C13	0–1	115	71	1500
	1–2	168	72	3400
PZ8	0–1	196	82	3000
	1–2	191	110	4000
S3	0–1	60	38	5100
	1–2	101	120	7000
	2–2.5	190	72	3200

**Table 2 plants-11-02250-t002:** Soil texture (sand, clay, silt) of the soil samples expressed as percentages.

Sample	Depth	% Sand	% Clay	% Silt	Sample	Depth	% Sand	% Clay	% Silt
**BH3**	0–1	86.1	4.81	9.10	**C41**	0–1	77.7	7.89	14.4
	1–2	89.8	3.74	6.51		1–2	85.0	5.53	9.43
**C05**	0–1	85.4	6.87	7.72	**PZ9**	0–1	94.7	1.56	3.73
	1–2	93.7	1.47	4.85		1–2	89.6	2.73	7.63
**PZ7**	0–1	90.8	3.23	6.01	**C13**	0–1	94.3	1.71	4.03
	1–2	86.0	5.18	8.78		1–2	91.9	2.08	6.03
**C08**	0–1	87.2	4.25	8.56	**PZ8**	0–1	87.8	6.28	5.92
	1–2	92.7	1.72	5.61		1–2	94.7	1.42	3.90

**Table 3 plants-11-02250-t003:** Composition of the four soils obtained by combining the sampled soils.

Composite Soils	Depth (m)	C > 12 (mg kg^−1^)
**B1** (BH3+CO8+PZ8)	0–1	3800
**B2** (CO5+PZ9+ PZ7+C13+C41)	0–1	1820
**B3** (BH3+PZ7+C13+C41)	1–2	3475
**B4** (CO5+PZ9+CO8+PZ8)	1–2	4800

**Table 4 plants-11-02250-t004:** Total microbial counts (expressed as colony forming units, CFU g^−1^) of the cultivable microorganisms detected in the different soil samples. The CFUs obtained are sufficient to indicate possible ongoing biodegradation activity.

Point	Depth	CFU g^−1^	Point	Depth	CFU g^−1^
**BH3**	0–1	4.0 × 10^6^	**C41**	0–1	1.0 × 10^8^
	1–2	4.0 × 10^6^		1–2	1.2 × 10^7^
**C05**	0–1	1.6 × 10^7^	**PZ9**	0–1	3.2 × 10^7^
	1–2	4.0 × 10^6^		1–2	1.6 × 10^7^
**PZ7**	0–1	4.0 × 10^7^	**C13**	0–1	8.0 × 10^6^
	1–2	1.6 × 10^7^		1–2	2.8 × 10^7^
**C08**	0–1	3.2 × 10^7^	**PZ8**	0–1	6.0 × 10^7^
	1–2	1.2 × 10^8^		1–2	6.0 × 10^7^
**S3**	0–2.5	4.0 × 10^7^			

**Table 5 plants-11-02250-t005:** Fresh weight (g) of plants (shoots and roots) grown on different soils. The reported value is the mean of the replicates with the standard deviations.

	Corn		Alfalfa		Lupine	
	shoots	roots	shoots	roots	shoots	Roots
B1	4.12 ± 0.42	4.93 ± 0.65	2.99 ± 0.61	2.29 ± 1.00	9.39 ± 2.55	3.07 ± 0.50
B2	4.43 ± 0.19	4.48 ± 0.59	2.61 ± 0.52	1.90 ± 0.50	8.95 ± 1.06	3.88 ± 1.08
B3	3.89 ± 0.46	3.60 ± 0.35	2.43 ± 1.39	0.57 ± 0.24	7.42 ± 1.64	2.64 ± 0.45
B4	3.34 ± 0.15	3.53 ± 0.26	2.63 ± 1.48	0.68 ± 0.06	6.71 ± 0.40	3.00 ± 0.48
CT	5.51 ± 0.63	7.02 ± 1.01	3.60 ± 0.43	2.84 ± 0.70	10.76 ± 0.85	3.64 ± 0.12

## Data Availability

Data are contained within the present article.

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
