# Peer review of "Nature-Based Solutions for Restoring an Agricultural Area Contaminated by an Oil Spill"

_plants, 2022, doi:10.3390/plants11172250_

Round 1

Reviewer 1 Report

I have carefully read your manuscript entitled: "Nature-Based Solutions to Restore an Agricultural Area Contaminated by an Oil Spill". Oil spills are a disaster to our environment that need to be addressed because can cause a disastrous effect quickly. That is why, in my opinion, the manuscript deals with a very important issue. The biodegradation percentages described by the authors are very promising. I only have minor comments. Please see them below.

- please, add information in the Abstract section, that the soil used in the experiment was taken from a polluted environment, not artificially contaminated - in my opinion, this is a great value of your study;

- line 54 - please describe what "near-zero discharge" is;

- lines 55-63 - it can be one paragraph;

- line 63 - please, give examples of these effects;

- line 264 - do the repetitions indicate the number of plants taken for testing (5)? or maybe the collected material was mixed, homogenized and divided into 5?

- lines 309 and 330 - please, add NA, L, LB description so that the potential reader does not have to look for explanations of the abbreviations in the text while looking at the figure;

- line 311 - please, define what biodegradation means (it is first time when you pointed this term);

- Figure 4, Table 5, Figure 6, Figure 7 - have you checked whether there are significant differences between the obtained results? it can make a difference to discuss;

- in my opinion, the conclusion section is too long - it should contain only the most important summary.

Best regards,

Reviewer

Reviewer 2 Report

Title: Nature-Based Solutions to Restore an Agricultural Area Contaminated by an Oil Spill

Ms No.: plants-1870576

The authors studied and reported about the Nature-Based Solutions to Restore an Agricultural Area Contaminated by an Oil Spill. The methodology was well organized and the results were properly discussed with other reported data. The work was suitable for the publication. However, before going to accept, few corrections need to be incorporated.

1.      Introduction needs to be improved.  

2.      Discussion part should be more precise

3.      Conclusion part requires revision

4.      Grammatical mistakes need to be fixed.

Reviewer 3 Report

Dear authors,

Thank you for your work!

This is an interesting paper on applied research, to restore the negative impact in an agricultural field contaminated by diesel oil, exploring two different approaches to bioremediation.

However, I suggested some improvements and my comments were included throughout the text. 

In general:

The introduction can be improved, including more detailed information and more references;

Some information is missing from the methodology, in order to support the defined objectives;

There is no clear separation between methodology and results; 

The discussion is poor and should be improved, e.g. it lacks comparisons with previously published studies;

The statistical analysis is referred (in line 300), but it is not included in the discussion of the results;

The conclusions are not totally supported by the obtained 

Best regards!

Round 2

Reviewer 2 Report

It can be accept in the current form